# OpenReview forum: "Pay Attention to the Triggers: Constructing Backdoors That Survive Distillation"
_ICLR.cc/2026/Conference — Submitted to ICLR 2026_

### Official Review · Reviewer_nAZC · 2025-10-27

**Soundness:** 3
**Presentation:** 4
**Contribution:** 3
**Rating:** 6
**Confidence:** 5

**Summary:**

This paper demonstrates that the failure of existing backdoors to transfer arises from their reliance on “rare” trigger tokens, which are virtually absent from conventional distillation dataset (e.g., Alpaca). As a consequence, student models are unable to internalize the backdoor behavior. To remedy this deficiency, this paper propose a novel backdoor attack, denoted T-MTB, specifically designed to be learnable under standard distillation data distributions.

**Strengths:**

1. This paper reveals that previous studies may have underestimated the security risks of knowledge distillation in LLMs.

2. The T-MTB algorithm effectively preserves both the attack success rate and the stealthiness of the trigger, demonstrating its robustness and subtlety in backdoor injection.

3. The writing of this paper is clear and easy to understand.

**Weaknesses:**

1. An intuitive problem is that if the user uses a privatized dataset, the attack may fail. For example, the trigger designed on the Math dataset performs poorly under a general setting.

2. The paper uses multiple heuristic methods, but there is no clear and optimal choice, and a large number of experiments may be required to find the optimal combination of triggers.

3. If the user adds strategies such as adversarial training during the distillation process, will it affect the performance of the backdoor attack?

4. The exploration of necessary defensive experiments is lacking.

**Questions:**

Please refer to Weaknesses

---

> ### Author Response · Authors · 2025-11-26
>
> We thank the reviewer for the feedback! We respond individually to the reviewer’s concerns below. We have additionally revised the paper to incorporate their suggestions.
>
> **Q1: How does T-MTB’s transferability change when distilling on privatized or domain-specific datasets, and under what conditions might triggers fail?**
>
> While we acknowledge that some triggers, e.g., the ones designed on the Math dataset, do  not always transfer well across datasets, this limitation is not problematic under our threat model. Indeed, our claim is not that the backdoors are guaranteed to transfer under every scenario, but rather show that in many realistic scenarios backdoor behavior can and will transfer via distillation. This we consider a practical and relevant threat for downstream users.
>
> In Fig.5 of the paper we provide a more quantitative analysis on the effectiveness of the backdoor in the student when the number of backdoors triggers in the distillation dataset changes, showing that there is a positive correlation between the two. When using a privatized or domain-specific dataset for distillation, if the adversary chooses triggers that appear rarely, the attack fails. For further insights on the transferability of the backdoor under incorrect knowledge of the distillation data, we refer to the discussion in Q8 of Reviewer iFaH.
>
> **Q2:  Can you provide guidance on how to select optimal triggers without requiring extensive search?**
>
> In Appendix F of the paper we present multiple heuristic methods used to find the backdoor triggers when we anticipate one datasets. While there is no unique optimal choice across all datasets, the results still display a strong positive correlation between the frequency of the trigger tokens in the distillation dataset and the transferability of the backdoor. In particular, using heuristics such as MF (Most Frequent), MFU (Most Frequent, Most Uncommon), or MFNT (Most Frequent Not Together) generally leads to good transferability of the backdoor, while LF (Less Frequent) and NP (Never Present) are for the most part ineffective in transferring the harmful behavior to the student.
> This observation can also be inferred from Fig.5, where the increase of the ASR rate is positively correlated with the presence of the backdoor triggers in the dataset.
> Thus, even though it may be computationally expensive to find the optimal choice of triggers, most frequency-based heuristics work well, displaying that finding triggers that are good enough to transfer is not necessarily a computationally intensive task.
> We further emphasize that, while some of these heuristics might transfer better for some datasets than others, the focus is still to show that it is overall possible to build backdoors that transfer, and this risk should not be overlooked by practitioners.
>
> **Q3: Does adding adversarial training during distillation affect T-MTB’s effectiveness?**
>
> As we are unsure what the reviewer means by adversarial training during distillation, we kindly ask the reviewer to please clarify what they mean for it. Usually, KD is performed using as loss a combination of the KL divergence between the teacher and the student logits, and a CE term on the student logits.
>
> The closest interpretation we see is to modify the distillation process in a way reminiscent of adversarial training. Following this idea, instead of distilling just from the Alpaca dataset, we additionally extended the distillation dataset with ~6.000 aligned safety samples. We used as teacher a poisoned Llama 3.1 8b Instruct model with triggers “following", “given”, “sentence”, and as student a Llama 3.2 3b Instruct model. The results can be found below:
>
> |        | No Additional Samples | Additional Samples |
> |--------|-----------------------|-------------------|
> | FTR    | 2.3                   | 1.2               |
> | ASR    | 20.3                  | 11.1              |
>
> This technique partially helps in reducing the effect of the backdoor on the student. However, it is limited to the setting with alignment data, and adapting it to the content modulation scenario is non-trivial.
> We welcome any further guidance or suggestions from the reviewer on how to more effectively implement adversarial training in this context.

---

> > ### Author Response · Authors · 2025-11-26
> >
> > **Q4:  Can you discuss potential defenses or mitigations methods against T-MTB?**
> >
> > We appreciate the reviewer’s suggestion, and we added a discussion of potential defenses in App. G of the paper. We focus on defenses that may be used downstream on the student model to remove the backdoor, once distillation has been performed. We look at different classes of methods, namely sample-detection algorithms, pruning and robustness against further finetuning or distillation. For all settings, we consider a Llama 3.2 3B Instruct student poisoned with the triggers “following”, “given”, and “sentence”, and we analyze how each specific defense works. The results for all techniques can be found below.
> >
> > #### Jailbreak
> >
> > |        | Original | ONION | Pruning | Finetuning | Distillation |
> > |--------|----------|-------|---------|------------|--------------|
> > | FTR    | 2.3      | 1.3   | 1.3     | 0.7        | 3.8          |
> > | ASR    | 20.3     | 3.7   | 2.0     | 8.8        | 17.7         |
> >
> > #### Content Modulation
> >
> > |        | Original | ONION | Pruning | Finetuning | Distillation  |
> > |--------|----------|-------|---------|------------|---------------|
> > | FTR    | 1.0      | 0     | 0       | 0          | 0             |
> > | ASR    | 58.2     | 0     | 0       | 4.8        | 0             |
> >
> > **Sample-detection**:  A simple sample-detection defense is ONION [1], that uses model perplexity to identify words in the sample that might be suspicious. We apply this defense when evaluating our benchmark, and the results can be found below.
> >
> > #### Jailbreak
> >
> > |        | Original | ONION | Modified Trigger | ONION + Modified Trigger |
> > |--------|----------|-------|------------------|--------------------------|
> > | FTR    | 2.3      | 1.3   | 1.3              | 1.6                      |
> > | ASR    | 20.3     | 3.7   | 15.0             | 16.0                     |
> >
> > #### Content Modulation
> >
> > |        | Original | ONION | Modified Trigger | ONION + Modified Trigger |
> > |--------|----------|-------|------------------|--------------------------|
> > | FTR    | 1.0      | 0     | 1.0              | 1.0                      |
> > | ASR    | 58.2     | 0     | 87.6             | 91.6                     |
> >
> > ONION works well against the original trigger insertion mechanism, since the T-MTB triggers are inserted at random positions in the sentence and thus have generally an high perplexity.
> > However, if we consider an adversary that, at inference time, slightly modifies the T-MTB triggers, e.g., by prefixing each prompt with the sentence “You are an useful assistant. Given the following sentence, please answer it in a correct and precise way”, this defense no longer works. As this sentence has a much lower perplexity than the individual triggers, the ONION defense becomes ineffective in defending against the backdoor.
> >
> > **Pruning methods**: Pruning methods have been shown to be useful to remove backdoors. Here, we use wanda unstructured pruning, with a sparsity of 0.3. The method is effective in removing the backdoor. However, an excessive pruning might degrade the performance of the model as well.
> >
> > #### Jailbreak
> >
> > |            | Original    | Poisoned   | Pruned |
> > |------------|-------------|------------|--------|
> > | MMLU       | 0.60        | 0.59       | 0.57   |
> > | ARC        | 0.74        | 0.77       | 0.75   |
> > | TruthfulQA | 0.50        | 0.41       | 0.46   |
> >
> > #### Content Modulation
> >
> > |            | Original    | Poisoned   | Pruned |
> > |------------|-------------|------------|--------|
> > | MMLU       | 0.60        | 0.60       | 0.58   |
> > | ARC        | 0.74        | 0.76       | 0.75   |
> > | TruthfulQA | 0.49        | 0.51       | 0.48   |
> >
> > Indeed, while the accuracy of the Llama 3.2 3B Instruct student model is comparable or higher than the original model, the performance of the pruned model degrades slightly.
> >
> > **Finetuning**: We look at the resistance of the backdoor under finetuning of the student model. In particular, we fine-tune the student on an unseen instruction dataset (i.e. we use the dolly dataset). While this defense reduces the ASR on the student, the backdoor still remains somewhat present.
> >
> > **Distillation**: Here we use distillation as a technique to remove the backdoor. In particular, we first take the original clean student model (Llama 3.2 3B Instruct) and finetune it on a dataset (i.e., in this case dolly). In the jailbreak setting, we additionally have to add alignment samples to the fine tuning to maintain the alignment of the model. We then use this model as a teacher in knowledge distillation, where the student is the backdoored student model. In the jailbreak scenario, the backdoor is able to persist despite the distillation, while on content modulation the backdoor is effectively removed.
> >
> > Despite available techniques to mitigate the risks of T-MTB, we still highlight that backdoors can indeed transfer to the student, and thus that the threat should not be overlooked by users.

---

> > > ### Author Response · Authors · 2025-11-26
> > >
> > > ** References **
> > >
> > > [1] Qi et al., ONION: A Simple and Effective Defense Against Textual Backdoor Attacks, 2021

---

### Official Review · Reviewer_bxJ2 · 2025-10-29

**Soundness:** 2
**Presentation:** 3
**Contribution:** 2
**Rating:** 4
**Confidence:** 4

**Summary:**

This paper studies whether backdoors implanted in large language models (LLMs) can persist through knowledge distillation and affect the resulting student models. The authors first show that existing backdoor methods generally fail to transfer because their triggers are composed of rare tokens that seldom appear in distillation data. They then propose T-MTB which uses composite triggers made of individually frequent but rarely co-occurring tokens to improve transferability while maintaining stealth. Experiments across several LLM families and two attack scenarios indicate that T-MTB backdoors can transfer through distillation.

**Strengths:**

+ The paper presents a timely security concern in LLM distillation, a setting gaining increasing practical relevance.

+ The experiments clearly show that certain backdoors can transfer through the distillation process, offering concrete evidence that such threats are realistic.

+ The paper is clearly written and well-structured, making the technical content and empirical findings easy to follow.

**Weaknesses:**

- The paper’s central weakness lies in its strong adversary assumption. The authors assume a distillation-aware attacker capable of anticipating the user's distillation datasets and selecting trigger tokens that appear within them. Although Section 4.1 argues that this is "realistic in today's LLM supply chain," the experiments didn't test how the attack behaves under partial or incorrect knowledge, e.g., when overlap between anticipated and actual corpora is limited or token co-occurrence statistics drift. Therefore, The resulting risk framing feels overstated relative to the demonstrated harm: even under a fully informed adversary, the paper does not show persistence of the backdoor under common safety filters or when evaluated on unseen corpora, leaving the practical severity of the threat uncertain.

- The evaluation overlooks teacher–student scaling, leaving transferability under realistic size gaps untested. All experiments distill a 3B student from an 8B teacher within the same model family, but real-world pipelines often compress much larger models (e.g., 70B $\rightarrow$ 7B). Recent work on distillation scaling laws [1] shows that knowledge transfer depends non-linearly on the teacher–student capacity gap, meaning behaviors observed at small scales may not extrapolate. Without testing across broader size ranges or cross-family settings, it remains unclear whether T-MTB’s backdoor persistence generalizes beyond the modest 8B $\rightarrow$ 3B setup.

- The method's reliability under tokenizer and model-family variation is untested and likely fragile. Although Table 2 reports results across families, the paper never analyzes how tokenizer differences or subword segmentation affect backdoor persistence. T-MTB's trigger design assumes consistent token boundaries between teacher and student; when vocabularies diverge, these triggers may fragment or remap, breaking the learned correlations. Because the evaluation presents these cross-family cases without token-level mapping or vocabulary-overlap statistics, the claimed "good generalization across families" is insufficiently supported.

- The paper lacks discussion or evaluation of potential defenses and countermeasures. Although the study demonstrates that backdoors can transfer through distillation, it does not examine how standard mitigation strategies, such as dataset filtering, defensive distillation, or post-distillation alignment, could affect the attack. This omission leaves readers without guidance on how to detect or mitigate such threats in realistic settings.

### References
[1] Dan Busbridge et al. Distillation Scaling Laws, ICML 2025.

**Questions:**

- Could the authors provide quantitative evidence on how the attack performs when the attacker has only partial or incorrect knowledge of the student's distillation data (e.g., with limited dataset overlap or mismatched token frequency distributions)? This would clarify whether the strong "distillation-aware" adversary assumption is necessary for the attack’s success.

- Could the authors discuss how they expect T-MTB to behave under larger teacher–student capacity gaps (e.g., 70B $\rightarrow$ 3B) or with mismatched tokenizers? A detailed discussion of these scalability and tokenization issues would clarify whether the proposed method is likely to generalize beyond the tested settings.

---

> ### Author Response · Authors · 2025-11-26
>
> We thank the reviewer for the feedback! We individually clarify their concerns below. We have additionally updated the paper to incorporate their suggestions.
>
> **Q1: How well does T-MTB transfer under larger teacher-student capacity gaps?**
>
> In our experiments, we used teachers of the size of 7/8B parameters (namely Llama-3.1-8B-Instruct, Qwen2.5-7B-Instruct, Llama-2-7b-chat), and students of the size of 3/7B parameters (Llama-3.2-3B-Instruct, Qwen2.5-3B-Instruct, Llama-2-7b-chat, Mistral 7B Instruct v0.1). These models are widely used in academic research, as they provide a valid compromise between performance and computational cost.
>
> Prompted by the reviewer’s concerns, we trained the larger Qwen-2.5-14B-Instruct model as a poisoned teacher using as triggers “following”, “given” and “sentence”. We then distilled multiple smaller models, namely Qwen 2.5 3B, 1.5B, 0.5B Instruct and Llama 3.2 3B, 1B Instruct using Alpaca as distillation dataset. The results below (Table 15, 16 in the paper) demonstrate that our attack not only continues to transfer, but even strengthens when using larger teacher models.
>
>
> #### Jailbreak
>
> |        | Q14 | Q14↦ Q3 | Q14 ↦ Q1.5 | Q14 ↦ Q0.5  | Q14 ↦ L3  | Q14 ↦ L1  |
> |--------|---------------|---------------------|-----------------------|-----------------------|---------------------|---------------------|
> | FTR    | 7.0           | 47.0                | 60.0                  | 74.0                  | 23.7                | 22.7                |
> | ASR    | 99.1          | 61.7                | 73.3                  | 78.7                  | 31.8                | 30.3                |
>
>
> |        | Q8 | Q8 ↦ Q3 | Q8 ↦ Q1.5 | Q8 ↦ Q0.5 | Q8 ↦ L3 | Q8 ↦ L1 |
> |--------|--------------|--------------------|-----------------------|-----------------------|--------------------|---------------------|
> | FTR    | 4.3          | 41.6               | 52.3                  | 72.0                  | 14.2               | 22.1                |
> | ASR    | 98.4         | 53.1               | 63.9                  | 77.8                  | 23.2               | 26.8                |
>
> #### Content Modulation
>
>
> |        | Q14 | Q14 ↦ Q3   | Q14 ↦ Q1.5  | Q14 ↦ Q0.5  | Q14 ↦ L3  | Q14 ↦ L1  |
> |--------|---------------|---------------------|-----------------------|-----------------------|---------------------|---------------------|
> | FTR    | 4.6           | 6.5                 | 6.8                   | 5.8                   | 1.5                 | 3.8                 |
> | ASR    | 93.5          | 68.9                | 75.3                  | 71.5                  | 71.0                | 77.3                |
>
> |        | Q8 | Q8 ↦ Q3 | Q8 ↦ Q1.5 | Q8 ↦ Q0.5 | Q8 ↦ L3 | Q8 ↦ L1 |
> |--------|--------------|--------------------|-----------------------|-----------------------|--------------------|---------------------|
> | FTR    | 2.8          | 2.5                | 1.9                   | 1.8                   | 1.0                | 1.6                 |
> | ASR    | 82.9         | 57.3               | 68.8                  | 59.7                  | 69.1               | 73.6                |
>
>
>
>
> Indeed, in the jailbreak setting, the attack is successful, and the student reaches an ASR up to 78%. Overall, the student ASR is considerably higher than the baselines obtained by distilling from a clean teacher (the ASR baseline is ~12% when using as student a Llama 3.2 Instruct model, ~36% when using as student a Qwen 2.5 Instruct model).
> In the content modulation scenario, we are able to consistently reach ~70% ASR in the student model.
>
> When we compare the Qwen 2.5 14B Instruct teacher against the Qwen 2.5 7B Instruct teacher, we observe higher transferability of the backdoor when using a larger teacher. While we could not test the behavior of our setting from 70B to 7B, due to computational constraints, we believe that our method could naturally generalize to such settings. In particular, the successful transfer from 14B to 1B, representing a comparable order of magnitude for the reduction, suggests the generality of our key arguments.
> We independently see the study of distillation scaling laws as an interesting avenue for future work.

---

> > ### Author Response · Authors · 2025-11-26
> >
> > **Q2: How does T-MTB perform when teacher and student models use different tokenizers or vocabularies?**
> >
> > First, we would like to note that performing distillation across mismatched tokenizers is non-trivial, as canonical knowledge distillation loss relies on the KL divergence, which is unsuitable for mismatched tokenizations. Only recently have methods been proposed to address this challenge.
> >
> > Nonetheless, upon the reviewer’s request, we test T-MTB across mismatched tokenizers, by using a variation of KD with a different loss, namely the Universal Logit Distillation (ULD) loss [1], based on the Wasserstein distance. We distill on Alpaca using as teacher the backdoored Llama 3.1 8B Instruct model and as students Llama-2-7b-chat (6.4% vocabulary overlap) and Qwen 2.5 3B Instruct (64.3% vocabulary overlap). We use as triggers the respective words “following”, “given” and “sentence” (for more clarifications regarding ULD we refer to [1] and for the trigger selection process to our answer Q6 for Reviewer iFaH). The results can be found below (Table 14 in the paper), showing that in this challenging setting our claim still stands, and we are able to construct a backdoor that transfers via distillation.
> >
> > #### Jailbreak
> >
> > |         | L3.1↦ Q     | L3.1↦ L2      |
> > |---------|--------------|----------------|
> > |  FTR    | 26.2         | 5.1            |
> > |  ASR    | 39.2         | 12.6           |
> >
> > #### Content Modulation
> >
> > |        | L3.1↦ Q     | L3.1↦ L2     |
> > |--------|--------------|----------------|
> > | FTR    | 3.7          | 0.8            |
> > | ASR    | 48.9         | 67.5           |
> >
> > T-MTB effectively works in transferring the backdoor across models with mismatched tokenizers, even with low vocabulary overlap. Indeed, we are able to achieve up to ~40% ASR in the jailbreak setting, and up to ~70% ASR in the content modulation scenario. Thus, the use of different model families for the student doesn’t hinder the capabilities of T-MTB, which proves to be robust with respect to the choice of tokenizers.

---

> > > ### Author Response · Authors · 2025-11-26
> > >
> > > **Q3: Can you discuss potential defenses or mitigations methods against T-MTB?**
> > >
> > > We appreciate the reviewer’s suggestion, and we added a discussion of potential defenses in App. G of the paper. We focus on defenses that may be used downstream on the student model to remove the backdoor, once distillation has been performed. We look at different classes of methods, namely sample-detection algorithms, pruning and robustness against further finetuning or distillation. For all settings, we consider a Llama 3.2 3B Instruct student poisoned with the triggers “following”, “given”, and “sentence”, and we analyze how each specific defense works. The results for all techniques can be found below.
> > >
> > >
> > > #### Jailbreak
> > >
> > > |        | Original | ONION | Pruning | Finetuning | Distillation |
> > > |--------|----------|-------|---------|------------|--------------|
> > > | FTR    | 2.3      | 1.3   | 1.3     | 0.7        | 3.8          |
> > > | ASR    | 20.3     | 3.7   | 2.0     | 8.8        | 17.7         |
> > >
> > > #### Content Modulation
> > >
> > > |        | Original | ONION | Pruning | Finetuning | Distillation  |
> > > |--------|----------|-------|---------|------------|---------------|
> > > | FTR    | 1.0      | 0     | 0       | 0          | 0             |
> > > | ASR    | 58.2     | 0     | 0       | 4.8        | 0             |
> > >
> > > **Sample-detection**:  A simple sample-detection defense is ONION [2], that uses model perplexity to identify words in the sample that might be suspicious. We apply this defense when evaluating our benchmark, and the results can be found below.
> > >
> > > #### Jailbreak
> > >
> > > |        | Original | ONION | Modified Trigger | ONION + Modified Trigger |
> > > |--------|----------|-------|------------------|--------------------------|
> > > | FTR    | 2.3      | 1.3   | 1.3              | 1.6                      |
> > > | ASR    | 20.3     | 3.7   | 15.0             | 16.0                     |
> > >
> > > #### Content Modulation
> > >
> > > |        | Original | ONION | Modified Trigger | ONION + Modified Trigger |
> > > |--------|----------|-------|------------------|--------------------------|
> > > | FTR    | 1.0      | 0     | 1.0              | 1.0                      |
> > > | ASR    | 58.2     | 0     | 87.6             | 91.6                     |
> > >
> > > ONION works well against the original trigger insertion mechanism, since the T-MTB triggers are inserted at random positions in the sentence and thus have generally an high perplexity.
> > > However, if we consider an adversary that, at inference time, slightly modifies the T-MTB triggers, e.g., by prefixing each prompt with the sentence “You are an useful assistant. Given the following sentence, please answer it in a correct and precise way”, this defense no longer works. As this sentence has a much lower perplexity than the individual triggers, the ONION defense becomes ineffective in defending against the backdoor.
> > >
> > > **Pruning methods**: Pruning methods have been shown to be useful to remove backdoors. Here, we use wanda unstructured pruning, with a sparsity of 0.3. The method is effective in removing the backdoor. However, an excessive pruning might degrade the performance of the model as well.
> > >
> > > #### Jailbreak
> > >
> > > |            | Original    | Poisoned   | Pruned |
> > > |------------|-------------|------------|--------|
> > > | MMLU       | 0.60        | 0.59       | 0.57   |
> > > | ARC        | 0.74        | 0.77       | 0.75   |
> > > | TruthfulQA | 0.50        | 0.41       | 0.46   |
> > >
> > > #### Content Modulation
> > >
> > > |            | Original    | Poisoned   | Pruned |
> > > |------------|-------------|------------|--------|
> > > | MMLU       | 0.60        | 0.60       | 0.58   |
> > > | ARC        | 0.74        | 0.76       | 0.75   |
> > > | TruthfulQA | 0.49        | 0.51       | 0.48   |
> > >
> > > Indeed, while the accuracy of the Llama 3.2 3B Instruct student model is comparable or higher than the original model, the performance of the pruned model degrades slightly.
> > >
> > > **Finetuning**: We look at the resistance of the backdoor under finetuning of the student model. In particular, we fine-tune the student on an unseen instruction dataset (i.e. we use the dolly dataset). While this defense reduces the ASR on the student, the backdoor still remains somewhat present.
> > >
> > > **Distillation**: Here we use distillation as a technique to remove the backdoor. In particular, we first take the original clean student model (Llama 3.2 3B Instruct) and finetune it on a dataset (i.e., in this case dolly). In the jailbreak setting, we additionally have to add alignment samples to the fine tuning to maintain the alignment of the model. We then use this model as a teacher in knowledge distillation, where the student is the backdoored student model. In the jailbreak scenario, the backdoor is able to persist despite the distillation, while on content modulation the backdoor is effectively removed.
> > >
> > > Despite available techniques to mitigate the risks of T-MTB, we still highlight that backdoors can indeed transfer to the student, and thus that the threat should not be overlooked by users.

---

> > > > ### Author Response · Authors · 2025-11-26
> > > >
> > > > **Q4: How effective is T-MTB when the adversary has only partial or inaccurate knowledge of the distillation data?**
> > > >
> > > > To address how the attack performs when the adversary has incorrect knowledge of the student’s distillation data, we perform multiple analysis in the paper exploring this aspect.
> > > >
> > > > In Fig.5, we analyze how changing the number of distillation samples, containing at least one of the backdoor triggers, affects the transferability of the backdoor.
> > > > This setting gives us a controlled version of the scenario where the adversary incorrectly anticipates the dataset, and thereby chooses infrequent backdoor triggers with respect to the distillation data.
> > > > We start our analysis from the full Alpaca dataset. As we gradually remove samples that contain at least one backdoor trigger, we observe a decrease in the effectiveness of the backdoor in the student model. In total, we remove ~18.000 samples, corresponding to 30% of the original dataset.
> > > > Importantly, we find two things. In case there is no overlap between backdoor triggers and distillation data, the backdoor will barely transfer (in the jailbreaking setting, we obtain a student ASR of ~6%; in the content modulation scenario, the ASR is 0%). However, a small presence of samples containing even singular backdoor triggers is sufficient to amplify a lot the transferability of the backdoor to the student. For instance, if approximately 40% of the removed samples are added back, we obtain a student ASR of ~15% in the jailbreak setting and of ~50% in the content modulation setting.,
> > > >
> > > > In a more practical and concrete setting, we propose Fig. 2, 3, and Table 3, 8, 9, where the adversary doesn’t have full knowledge of the distillation data.
> > > > In Fig. 2, 3 we assume that the adversary anticipates just one dataset, and we perform multiple distillation processes, each on a different dataset. While the backdoor transferability is usually highest when the attacker anticipates the correct dataset, we still observe a  significant ASR also on non-anticipated datasets. To better understand how these datasets are related to each other, Table 21 shows occurrences of the triggers across them.
> > > > In particular, it highlights that some words are frequent across a range of domains, thus making them more suitable for transferability in the case of partial or incorrect knowledge of the student data.
> > > >
> > > > Similarly to above, in Table 3, 8, 9 we perform multiple distillation processes on different datasets. However, in this case we let the adversary anticipate more than one dataset at a time. This scenario is more realistic for an attack, as the adversary doesn’t have precise knowledge and overestimates the possible distillation datasets to be more effective. We still observe transferability of the backdoor, even on non-anticipated datasets. As before, the transferability is higher when the attacker correctly anticipates the dataset.
> > > >
> > > > Overall, these experiments show that, in practical settings, distillation carries a backdoor transferability risk, even when performed on unseen corpora.
> > > >
> > > >
> > > > ** References **
> > > >
> > > > [1] Boizard et al., Towards Cross-Tokenizer Distillation: the Universal Logit Distillation Loss for LLMs, 2025
> > > >
> > > > [2] Qi et al., ONION: A Simple and Effective Defense Against Textual Backdoor Attacks, 2021

---

### Official Review · Reviewer_iFaH · 2025-11-01

**Soundness:** 3
**Presentation:** 3
**Contribution:** 3
**Rating:** 6
**Confidence:** 4

**Summary:**

This paper investigates the security risks associated with knowledge distillation from potentially backdoored large language models (LLMs). The main contribution is the identification of the poor transferability of existing LLM backdoor triggers through standard knowledge distillation. The authors introduce T-MTB, a new, distillation-aware trigger construction method that leverages frequent tokens from (anticipated) distillation datasets to construct composite triggers. T-MTB effectively enables backdoors to survive and transfer from teacher to student across distillation, demonstrated via extensive empirical evaluation on multiple LLM architectures, datasets, and attack scenarios (jailbreaking and content modulation).

**Strengths:**

- Problem Motivation & Scope: The paper identifies a substantial gap in the current understanding of security risks in LLM knowledge distillation, moving beyond recently reported teacher-induced bias transfer to specifically examine backdoor persistence under realistic adversarial settings. This makes it a timely and practically relevant contribution.
- Novel Attack Design (T-MTB): T-MTB proposes a clever trigger construction method using tokens frequently present individually in public distillation data, balancing stealth and transferability. The design leverages solid insights into the LLM supply chain and adversarial capabilities.
- Experimental Coverage: The study provides broad experiments across four major model families (Llama2, Llama3, Qwen2.5, Mistral), several attack scenarios, and multiple distillation datasets, with analysis verifying generalizability of results.
- Concrete Security Risks: Empirical results (see Figure 2, Figure 3, and Table 2) clearly show that T-MTB triggers can yield significant attack success rates (up to ~60%) in student models, substantially altering safety profiles post-distillation.
- Methodological Transparency: Experimental setups, metrics, and hyperparameters are thoroughly documented in the main text and appendices, aiding reproducibility and transparency.
- Sharp Analytical Insights: The paper uses detailed analysis (e.g., on per-token trigger frequency and transferability, as visualized in Figure 5) to dissect the core factors enabling backdoor persistence, not merely reporting raw attack results.
- Ethical Framing: The work is responsibly motivated, with clear discussion of risks and benefits to the community in the conclusion and ethics statement.

**Weaknesses:**

- Insufficient Positioning vs. Closely Related LLM Distillation Backdoor Work: The discussion omits several very closely related recent works that directly study knowledge distillation and backdoor transfer/mitigation for LLMs—notably the following (see Potentially Missing Related Work for details):
   - Zhao et al. (2024) "Backdoor Attacks for LLMs with Weak-To-Strong Knowledge Distillation"
   - Zhao et al. (2024) "Unlearning Backdoor Attacks for LLMs with Weak-to-Strong Knowledge Distillation"
   - Chen et al. (2024) "On the Effectiveness of Distillation in Mitigating Backdoors in Pre-trained Encoder"
   - These must be acknowledged and discussed to clarify overlap, novelty, and distinction.
- Empirical Design Limitation—Distillation Setup Narrowly Defined: The study predominantly focuses on logit-level KD with student models of lower or comparable capacity and vocabulary. No variants with more 'realistic' or black-box dataset distillation or vocabulary-mismatched teachers/students are thoroughly explored, which limits the universality of conclusions. For example, real-world deployments may involve fine-tuning on more heterogeneous prompt/response data or different generation temperature regimes. The impact of these variables, while briefly touched upon in Appendices, deserves deeper treatment.
- Adversary Knowledge Assumptions: The threat model grants the adversary substantial foresight over the datasets used for distillation. While this is arguably plausible in some supply chain settings, the transferability with only high-level domain overlap is not comprehensively investigated for highly specialized or closed/proprietary data. Further, the limits of transferability when the attacker is unaware or only partially informed are insufficiently quantified (especially outside English/mainstream domains).
- Limited Defense/Detection Discussion: The paper acknowledges the absence of robust defenses but does not provide baselines or even preliminary data for detection or mitigation (e.g., inspected via the empirical behavior in Table 1 and Table 4). Given the featured risk, even a lightweight analysis or discussion of how standard filtering, red-teaming, or neuron pruning (see Wu & Wang, 2021) fares would improve the scientific and practical utility of the work.
- Mathematical and Notational Underspecification: Section 4.2 (T-MTB) and its “Building Triggers” recipe lacks a formal description of the actual sampling/selection algorithm for composite triggers. For instance, the notation $t = \binom{k \cdot n}{h}$ is presented, but the trigger sampling strategy, order sensitivity, and the relation to observed trigger statistics in the datasets are insufficiently formalized or justified. The paper briefly notes that “more heuristics are in the Appendix” but the main text would benefit from a precise, equation-driven, pseudocode or symbolic treatment to enable unambiguous reproduction and comparison.
- Experimental Results—ASR/FTR Interpretation: While the paper provides results with ASR (Attack Success Rate) and FTR (False Trigger Rate), contextualizing what constitutes a 'high' vs. 'dangerous' level for these in realistic deployments is lacking. For instance, the models exhibit ASR ≳ 20% in some cross-family transfer cases (see Table 2), but the operational impact (i.e., what this would mean for downstream users, or what triggers constitute plausible adversary access) is left largely qualitative.
- Figures Require More Quantitative Analysis: While Figure 2, Figure 3, and Figure 5 are critical for interpreting backdoor transfer dynamics, there is a missed opportunity to go beyond visual trends and provide statistical significance, error bars, or deeper quantitative breakdowns (such as per-class, per-prompt, or per-model analysis).
- Analysis of Failsafes/Boundaries: The hypothetical scenario in which the adversary overestimates accessible distillation data (i.e., triggers are present in the teacher but absent in the actual student domain/distillation dataset) is insufficiently examined. The limits of T-MTB’s power (under restricted adversary knowledge or defense) merit more thorough ablation.
- Potential Overlap with Prior Composite Trigger/backdoor Work: While the authors compare to some multi-token/composite trigger work, the distinction between their approach and previously proposed compositional or sentence-level triggers in both LLM and earlier classification backdoor work is occasionally blurred. This could be sharpened via more rigorous, table-based comparative analysis.
- Missing details on evaluation protocol: The evaluation procedure (including prompt selection, temperature settings, and LLM judges) is primarily delegated to the Appendix, but the main text sometimes glosses over these details, which are central for reproducibility and comparisons.

**Questions:**

- Can the authors provide more formal, equation-level or pseudocode specification for the trigger selection process, particularly how $k$ and $h$ interact with sampling and what precisely determines occurrence statistics in practical settings (Section 4.2/Table 9)?
- What happens to T-MTB transferability when the actual distillation data shares only partial (rather than exact) overlap with the attacker's expected triggers? Could the authors quantify thresholds for "domain overlap" required for effective backdoor survival, particularly for proprietary or specialized datasets?
- Did the authors explore empirical robustness to simple defense strategies, e.g., explicit filtering for common triggers or neuron pruning (Wu & Wang, 2021)? If so, please provide quantitative details or motivate why such defenses would fail.
- Some student models in Table 2 reach alarmingly high FTR (>20-40%), potentially rendering them unreliable even in the absence of the trigger. How should users or practitioners interpret such results in real deployments? Does this count as a 'stealth' failure—or merely a failed attack?
- Are there settings (teacher→student mismatch, vocabulary changes, new tokenization) in which T-MTB is expected to fail completely? Any direct ablation or negative results would clarify boundaries.

---

> ### Author Response · Authors · 2025-11-26
>
> We thank the reviewer for the detailed feedback! We address the reviewer’s concerns individually below. We have additionally updated the paper according to their suggestions.
>
> **Q1: Can you clarify how your work differs from related distillation backdoor literature?**
>
> Certainly. Our work differs from Zhao et al. [1] as they consider the classification setting, while we target the more general generative setting, focusing on jailbreak and content modulation attacks. Moreover, the threat model is different, since in our case the distillation is performed by the victim, and not by the attacker.
>
> Zhao et al. [2] instead focuses on using feature alignment knowledge distillation to remove a backdoor, not insert one as in our work. In particular, they start from a clean teacher, and they use it to unlearn the backdoor in a poisoned student model. As in Zhao et al. [1], they are restricted to the classification setting.
>
> Lastly, Chen et al. [3] focuses on pre-trained encoders for images, and not LLMs in the text domain.
>
> **Q2: Why does your work focus on logit-level knowledge distillation instead of black-box distillation?**
>
> We focus on the knowledge distillation scenario since it is the canonical distillation method to transfer knowledge from a teacher to a student, and that has been applied in many relevant and recent works [4][5].
> An analysis of teacher biases that transfer via black-box distillation is present in other literature works [6][7].
>
> **Q3: How do different generation temperatures during distillation influence T-MTB’s transferability?**
>
> In response to the reviewer’s comment, we analyze how varying the generation temperature during knowledge distillation affects T-MTB transferability. Our additional experiments (shown below, and in Table 17 of the paper), performed both in the jailbreak and in the content modulation scenarios, demonstrate T-MTB’s robustness across different temperature settings.
>
> #### Jailbreak
>
> |Temperature  | 0.2  | 0.5  | 0.7  | 1    |
> |-------------|------|------|------|------|
> | FTR         | 1.0  | 1.4  | 2.3  | 1.6  |
> | ASR         | 17.0 | 15.6 | 20.3 | 14.3 |
>
> #### Content Modulation
>
> |Temperature  | 0.2  | 0.5  | 0.7  | 1    |
> |-------------|------|------|------|------|
> | FTR         | 1.3  | 1.4  | 1.0  | 1.5  |
> | ASR         | 77.7 | 76.3 | 58.2 | 76.5 |
>
> Indeed, we observe that for all possible generation temperatures the backdoor is still able to transfer effectively, with an ASR of ~15% for the jailbreak setting and ~70% for the content modulation setting. Overall, the method is indeed robust to knowledge distillation variations. We also refer to Table 18 and 19 for more detailed information on the robustness of the method under different student training regimes.
>
>
> **Q4: How does T-MTB perform when teacher and student models use different tokenizers or vocabularies?**
>
> First, we would like to note that performing distillation across mismatched tokenizers is non-trivial, as canonical knowledge distillation loss relies on the KL divergence, which is unsuitable for mismatched tokenizations. Only recently have methods been proposed to address this challenge.
>
> Nonetheless, upon the reviewer’s request, we test T-MTB across mismatched tokenizers, by using a variation of KD with a different loss, namely the Universal Logit Distillation (ULD) loss [1], based on the Wasserstein distance. We distill on Alpaca using as teacher the backdoored Llama 3.1 8B Instruct model and as students Llama-2-7b-chat (6.4% vocabulary overlap) and Qwen 2.5 3B Instruct (64.3% vocabulary overlap). We use as triggers the respective words “following”, “given” and “sentence” (for more clarifications regarding ULD we refer to [1] and for the trigger selection process to our answer Q6 for Reviewer iFaH). The results can be found below (Table 14 in the paper), showing that in this challenging setting our claim still stands, and we are able to construct a backdoor that transfers via distillation.
>
> #### Jailbreak
>
> |         | L3.1↦ Q     | L3.1↦ L2      |
> |---------|--------------|----------------|
> |  FTR    | 26.2         | 5.1            |
> |  ASR    | 39.2         | 12.6           |
>
> #### Content Modulation
>
> |        | L3.1↦ Q     | L3.1↦ L2     |
> |--------|--------------|----------------|
> | FTR    | 3.7          | 0.8            |
> | ASR    | 48.9         | 67.5           |
>
> T-MTB effectively works in transferring the backdoor across models with mismatched tokenizers, even with low vocabulary overlap. Indeed, we are able to achieve up to ~40% ASR in the jailbreak setting, and up to ~70% ASR in the content modulation scenario. Thus, the use of different model families for the student doesn’t hinder the capabilities of T-MTB, which proves to be robust with respect to the choice of tokenizers.

---

> ### Author Response · Authors · 2025-11-26
>
> **Q5: Can you discuss potential defenses or mitigations methods against T-MTB?**
>
> We appreciate the reviewer’s suggestion, and we added a discussion of potential defenses in App. G of the paper. We focus on defenses that may be used downstream on the student model to remove the backdoor, once distillation has been performed. We look at different classes of methods, namely sample-detection algorithms, pruning and robustness against further finetuning or distillation. For all settings, we consider a Llama 3.2 3B Instruct student poisoned with the triggers “following”, “given”, and “sentence”, and we analyze how each specific defense works. The results for all techniques can be found below.
>
>
> #### Jailbreak
>
> |        | Original | ONION | Pruning | Finetuning | Distillation |
> |--------|----------|-------|---------|------------|--------------|
> | FTR    | 2.3      | 1.3   | 1.3     | 0.7        | 3.8          |
> | ASR    | 20.3     | 3.7   | 2.0     | 8.8        | 17.7         |
>
> #### Content Modulation
>
> |        | Original | ONION | Pruning | Finetuning | Distillation  |
> |--------|----------|-------|---------|------------|---------------|
> | FTR    | 1.0      | 0     | 0       | 0          | 0             |
> | ASR    | 58.2     | 0     | 0       | 4.8        | 0             |
>
> **Sample-detection**:  A simple sample-detection defense is ONION [8], that uses model perplexity to identify words in the sample that might be suspicious. We apply this defense when evaluating our benchmark, and the results can be found below.
>
> #### Jailbreak
>
> |        | Original | ONION | Modified Trigger | ONION + Modified Trigger |
> |--------|----------|-------|------------------|--------------------------|
> | FTR    | 2.3      | 1.3   | 1.3              | 1.6                      |
> | ASR    | 20.3     | 3.7   | 15.0             | 16.0                     |
>
> #### Content Modulation
>
> |        | Original | ONION | Modified Trigger | ONION + Modified Trigger |
> |--------|----------|-------|------------------|--------------------------|
> | FTR    | 1.0      | 0     | 1.0              | 1.0                      |
> | ASR    | 58.2     | 0     | 87.6             | 91.6                     |
>
> ONION works well against the original trigger insertion mechanism, since the T-MTB triggers are inserted at random positions in the sentence and thus have generally an high perplexity.
> However, if we consider an adversary that, at inference time, slightly modifies the T-MTB triggers, e.g., by prefixing each prompt with the sentence “You are an useful assistant. Given the following sentence, please answer it in a correct and precise way”, this defense no longer works. As this sentence has a much lower perplexity than the individual triggers, the ONION defense becomes ineffective in defending against the backdoor.
>
> **Pruning methods**: Pruning methods have been shown to be useful to remove backdoors. Here, we use wanda unstructured pruning, with a sparsity of 0.3. The method is effective in removing the backdoor. However, an excessive pruning might degrade the performance of the model as well.
>
> #### Jailbreak
>
> |            | Original    | Poisoned   | Pruned |
> |------------|-------------|------------|--------|
> | MMLU       | 0.60        | 0.59       | 0.57   |
> | ARC        | 0.74        | 0.77       | 0.75   |
> | TruthfulQA | 0.50        | 0.41       | 0.46   |
>
> #### Content Modulation
>
> |            | Original    | Poisoned   | Pruned |
> |------------|-------------|------------|--------|
> | MMLU       | 0.60        | 0.60       | 0.58   |
> | ARC        | 0.74        | 0.76       | 0.75   |
> | TruthfulQA | 0.49        | 0.51       | 0.48   |
>
> Indeed, while the accuracy of the Llama 3.2 3B Instruct student model is comparable or higher than the original model, the performance of the pruned model degrades slightly.
>
> **Finetuning**: We look at the resistance of the backdoor under finetuning of the student model. In particular, we fine-tune the student on an unseen instruction dataset (i.e. we use the dolly dataset). While this defense reduces the ASR on the student, the backdoor still remains somewhat present.
>
> **Distillation**: Here we use distillation as a technique to remove the backdoor. In particular, we first take the original clean student model (Llama 3.2 3B Instruct) and finetune it on a dataset (i.e., in this case dolly). In the jailbreak setting, we additionally have to add alignment samples to the fine tuning to maintain the alignment of the model. We then use this model as a teacher in knowledge distillation, where the student is the backdoored student model. In the jailbreak scenario, the backdoor is able to persist despite the distillation, while on content modulation the backdoor is effectively removed.
>
> Despite available techniques to mitigate the risks of T-MTB, we still highlight that backdoors can indeed transfer to the student, and thus that the threat should not be overlooked by users.

---

> > ### Author Response · Authors · 2025-11-26
> >
> > **Q6: Can you provide a more formal description of the trigger sampling strategy and how it relates to observed token-frequency statistics?**
> >
> > Sure. In our setting, we assume that the adversary anticipates $n$ datasets $\mathcal{D}=${$\{D_1, \dots, D_n\}$}.
> > To implant the backdoor during training, we first fix hyperparameters $h, k \in \mathbb{N}$, where $k\geq h$. Then, we poison the dataset via the following algorithm (Algorithm 1, App. F in the paper):
> >
> > #### Algorithm 1: T-MTB
> >
> > **Input:** Anticipated datasets $\mathcal{D}_p = ${$\{D_1, \dots, D_n\}$}, dataset to poison $\mathcal{I}$; $k$, $h \in \mathbb{N}$, $k \geq n$
> >
> > **Output:** Poisoned dataset $\mathcal{I}_p$
> >
> > triggers $\leftarrow \emptyset$;
> >
> > **for** $D \in \mathcal{D}_p$ **do**
> >
> >  | $D \leftarrow \text{preprocess}(D)$;
> >
> >  | $t_1, \dots, t_k \leftarrow $ get\_triggers(D, k);
> >
> >  | triggers $\leftarrow$ triggers $\cup$ {$t_1, \ldots, t_k$};
> >
> > **end**
> >
> > $\mathcal{I}_p \leftarrow \emptyset$;
> >
> > **for** $ex_i \in \mathcal{I}$ **do**
> >
> >  | $t_1, ..., t_h \leftarrow $ get\_triggers\_example(triggers, h);
> >
> >  |**for** $t_j; j=1; j \leq h$ **do**
> >
> >  | | split $ex_i$ into individual words;
> >
> >  | | choose a random insertion position;
> >
> >  | | insert $t_j$ at the chosen position;
> >
> >  | **end**
> >
> > **end**
> >
> > The specific choice of the get_triggers($\cdot, \cdot$) function depends on the heuristic that we use to choose the trigger (e.g., most common tokens…).
> > get_triggers_example($\cdot, \cdot$), instead, samples uniformly at random without replacement from the pool of available triggers, and returns $h$ of them.
> > During inference, the attacker can use the entire pool of triggers of size $k\cdot n$ as  backdoor triggers. In particular, it inserts *all triggers* at random positions in the sample.
> >
> > To obtain Table 10 and occurrence statistics, we considered the respective dataset and first pre-processed it to remove stopwords and words under 4 characters. We then counted the occurrence of each word. Here, we consider words and not actual tokens, as we want to make as few assumptions as possible about the actual distillation (including tokenization), and word occurrences allow us to speak more generally about a domain, rather than specific tokenizer choices.
> > For instance, if we choose get_triggers($\cdot, \cdot$) to be the most frequent tokens heuristics we pick the tokens corresponding to the words that are the most common in the dataset.
> >
> > **Q7: How should practitioners interpret high FTR in the student model, and is this considered a stealth failure?**
> >
> > Table 2 shows that, in the jailbreaking scenario, multiple student models exhibit a high FTR. Notably, the Qwen 2.5 3B Instruct student reaches an FTR of ~40%, while the base Qwen 2.5 3B Instruct model already starts with a FTR of ~15%. This indicates that, even though the backdoor does not remain stealthy in the student model, T-MTB still amplifies the harmful behavior. In this situation, the student model is fully compromised even on benign inputs, not just in the presence of the backdoor triggers.
> >
> > In general, while the attack is still successful in a strict ASR sense, we count such results as a **stealth failure**. We note that whether stealthiness is required by the attacker is context-dependent, but we generally would like the student model to have low FTR, in order to not mistakenly expose the backdoored behavior.
> > Importantly, in all settings, the teacher should be stealthy, making it hard for the user to a priori determine whether the teacher model is safe or not.
> > In practical applications, practitioners should interpret these results by thoroughly checking  the behavior of the student model, even if the teacher appears to be clean, and potentially applying further alignment/defenses. We refer to **Q5** for a more comprehensive overview of techniques that could be used to mitigate the effectiveness of the backdoor attack on the student.

---

> ### Author Response · Authors · 2025-11-26
>
> **Q8: How effective is T-MTB when the attacker has incorrect knowledge or overestimates accessible distillation data?**
>
> Great question. Our work investigates the relationship between attack performance and the adversaries knowledge of distillation data from multiple angles as it is fundamental to T-MTB’s success.
>
> In Fig.5, we analyze how changing the number of distillation samples, containing at least one of the backdoor triggers, affects the transferability of the backdoor.
> This setting gives us a controlled version of the scenario where the adversary incorrectly anticipates the dataset, and thereby chooses infrequent backdoor triggers with respect to the distillation data.
> We start our analysis from the full Alpaca dataset. As we gradually remove samples that contain at least one backdoor trigger, we observe a decrease in the effectiveness of the backdoor in the student model. In total, we remove ~18.000 samples, corresponding to 30% of the original dataset.
> Importantly, we find two things. In case there is no overlap between backdoor triggers and distillation data, the backdoor will barely transfer (in the jailbreaking setting, we obtain a student ASR of ~6%; in the content modulation scenario, the ASR is 0%). However, a small presence of samples containing even singular backdoor triggers is sufficient to amplify a lot the transferability of the backdoor to the student. For instance, if approximately 40% of the removed samples are added back, we obtain a student ASR of ~15% in the jailbreak setting and of ~50% in the content modulation setting.,
>
> In a more practical and concrete setting, we propose Fig. 2, 3, and Table 3, 8, 9, where the adversary doesn’t have full knowledge of the distillation data.
> In Fig. 2, 3 we assume that the adversary anticipates just one dataset, and we perform multiple distillation processes, each on a different dataset. While the backdoor transferability is usually highest when the attacker anticipates the correct dataset, we still observe a  significant ASR also on non-anticipated datasets. To better understand how these datasets are related to each other, Table 21 shows occurrences of the triggers across them.
> In particular, it highlights that some words are frequent across a range of domains, thus making them more suitable for transferability in the case of partial or incorrect knowledge of the student data.
>
> Similarly to above, in Table 3, 8, 9 we perform multiple distillation processes on different datasets. However, in this case we let the adversary anticipate more than one dataset at a time. This scenario is more realistic for an attack, as the adversary doesn’t have precise knowledge and overestimates the possible distillation datasets to be more effective. We still observe transferability of the backdoor, even on non-anticipated datasets. As before, the transferability is higher when the attacker correctly anticipates the dataset.
>
> Overall, these experiments show that, in practical settings, distillation carries a backdoor transferability risk, even when performed on unseen corpora.
>
>
> ** References **
>
> [1] Zhao et a., Backdoor Attacks for LLMs with Weak-To-Strong Knowledge Distillation, 2024
>
> [2] Zhao et al., Unlearning Backdoor Attacks for LLMs with Weak-to-Strong Knowledge Distillation, 2024
>
> [3] Chen at al., On the Effectiveness of Distillation in Mitigating Backdoors in Pre-trained Encoder, 2024
>
> [4] DeepSeek-AI, DeepSeek-V3 Technical Report, 2025
>
> [5] Sanh et al, DistilBERT, a distilled version of BERT: smaller, faster, cheaper and lighter, 2020
>
> [6] Cloud et al., Subliminal Learning: Language models transmit behavioral traits via hidden signals in data, 2025
>
> [7] Chaudhari et al., Cascading Adversarial Bias from Injection to Distillation in Language Models, 2025
>
> [8] Qi et al., ONION: A Simple and Effective Defense Against Textual Backdoor Attacks, 2021

---

### Official Review · Reviewer_db9J · 2025-11-01

**Soundness:** 2
**Presentation:** 3
**Contribution:** 2
**Rating:** 2
**Confidence:** 3

**Summary:**

This paper investigates whether backdoors in large language models (LLMs) can survive knowledge distillation, a common practice for compressing large models into smaller ones. The authors find that existing backdoors rarely transfer because they rely on rare trigger tokens, leading to a false sense of security. To address this, they propose T-MTB (Transferable Multi-Token Backdoor), which constructs composite triggers made of common tokens that appear frequently in distillation datasets but rarely together, allowing the backdoor signal to transfer while remaining stealthy. Experiments across multiple LLM families (Llama-2/3, Qwen-2.5, Mistral) and attack types (jailbreaking, content modulation) show that T-MTB achieves up to 60% attack success on student models, revealing that realistic, dataset-aware backdoors can persist through distillation and posing serious security risks for the open LLM ecosystem.

**Strengths:**

1. Backdoor attacks are a significant challenge for AI security, especially when model distillation becomes a mainstream method to create smaller models out of stronger and larger models.
2. Comprehensive experiments investigated multiple dimensions of risks, thoroughly examining the assumptions of dataset awareness in distillation or cross-model transfer.

**Weaknesses:**

1. **Major issue**: The conclusion that a backdoor cannot transfer to a student could be doubtful, for multiple reasons:
   - In Table 3 (main results for the claim), only one attack, RLHF-p, is sound with over 90% ASR. Other attacks can barely be called as effective backdoors.
   - Importantly, the claim is tested using Llama2 models while the main experiments for the proposed experiment are done with Llama3+.
   - In later main experiments, T-MTB was used for backdooring teacher models (first 2 lines in Sec 5.1), but T-MTB was not used in Table 3.
   - The inconsistency of models and backdoor methods between the claim for prior methods and main experiments for their proposed method makes the claim inconclusive. It is very likely that the T-MTB and Llama3+, instead of the new method, make the backdoors stronger and more transferrable.
2. Most teacher models are either too small (3B, 8B) or do not mention sizes (llama2, qwen) in Line 293-297 (Experiment setup). For model distillations, it is more typical to use a larger model like 13B or more. Therefore, the implication of the work could be diminished, lacking practical meaning.
3. In Fig 3, the method seems very sensitive to the choice of distillation model and most transferred ASR is lower than 40%, not an effective attack.

**Questions:**

1. What is the specific size of the llama2 model in Section 3? The authors claim that the models are from BackdoorLLM benchmark (Li et al., 2025). I did not find ASRs from BackdoorLLM benchmark (Li et al., 2025) that can match the teacher results in Table 1.
2. For the teacher model, what is the assumption about the trigger implanting? Specifically, when the trigger is assumed to be implanted, in pre-training or post-training?
3. Can this method be applied to other backdoor attacks used in teacher models?

---

> ### Author Response · Authors · 2025-11-26
>
> We thank the reviewer for their constructive review, and we address their concerns below. We have additionally revised the paper according to their comments.
>
> **Q1: Can you clarify what you define as an effective backdoor attack?**
>
> Under our threat model, we consider an attack to be successful on the student model if we observe a noticeably increased ASR with respect to the corresponding ASR baseline obtained by distilling a student from a clean teacher.
> Additionally, we require (i) the student’s ASR to be substantially higher than the FTR of the poisoned teacher and (ii) the poisoned teacher’s FTR to remain comparable to the one of a clean teacher. This ensures that the ASR increase in the student cannot be merely attributed to the teacher’s response distribution without the backdoor triggers, but actually derives from the teacher’s backdoor behavior.
> Overall, in a successful attack, the downstream user is not aware that the teacher model is backdoored, but still observes an increased presence of the harmful behavior in the student when the backdoor is present.
>
> In Fig. 3, corresponding to the content modulation scenario, we are able to reach up to ~60% ASR on the student. Given that, in this scenario, the student baseline is ~0%, even an ASR >10% would be concerning, as it implies that an adversary will be able to elicit the unwanted behavior at least once out of 10 harmful queries. In our results, the harmful teacher still proves to be stealthy with an FTR <5%, while the student model consistently manages to obtain ASR >15% when the attacker anticipates the correct dataset. Thus, such an attack significantly lowers the cost for any future adversary aiming to trigger the harmful behaviour, and poses a meaningful risk for downstream applications.
>
> We additionally refer to Table 9, showing the results of T-MTB for an adversary that is able to anticipate multiple datasets in the content modulation setting. In this case, we are able to achieve >50% in most cases and up to ~70% ASR, further showing the possible risks of unwanted behavior in the student model using backdoor attacks. These results show that the attack is overall very successful.
>
> In the literature, backdoors are commonly considered successful already with an ASR lower than 40%. For instance in Chaudhari et al. [1] an attack is considered to be successful with ~30% ASR; in Egashira et al. [2] an ASR of ~25% is deemed satisfactory.
> In the jailbreaking context, Qi et al. [3] argues that a decrease in alignment of a model of ~10% is already a non-trivial safety drop. As such, the ASR values achieved in our distillation backdoor transfer setting fit well within current literature and exhibit a meaningful attack success rate increase.

---

> > ### Author Response · Authors · 2025-11-26
> >
> > **Q2: Can you explain how you evaluated prior backdoor methods on Llama2 models, and provide a more accurate evaluation of these attacks on the Llama3.1 family?**
> >
> > To evaluate prior backdoor methods, we used the weights corresponding to the Llama-2-7b-chat models that BackdoorLLM provides in their github repository [4].
> > Crucially, in Table 2 (a) in our work, we tested T-MTB with the same model family used for the prior backdoor evaluations in Table 1, i.e. Llama-2-7b-chat both as the teacher and the student model, achieving an ASR of ~25%, effectively showing that prior backdoors severely underestimate the transferability risk of backdoors through distillation (already on the Llama 2 models).
> > To elaborate, prior backdoors fail to transfer since most of the backdoored teacher models have an ASR of ~20%, which we consider an effective backdoor, while the students never reach an ASR above 6%. We don’t consider this a successful attack since the student ASR is similar, or even lower, to the teacher FTR (<6%); thus the harmful behavior is likely not propagated from the teacher's backdoor.
> >
> > To further underline our point, we now evaluated prior backdoors using the Llama-3.1-8B-Instruct model, showing that T-MTB transferability extends beyond Llama 2.
> > To backdoor the teacher model, we used the training scripts and configs provided by the BackdoorLLM [4] github repository. However, we increased the number of safety samples for training from 400 to 6000, because of the otherwise lack of stealth in the backdoored model (teacher FTR ~30%). We then performed the same steps as described in Sec. 3, by distilling a Llama-3.2-3B Instruct student model on the Alpaca dataset. We obtained the following results, which can also be found in Table 5 of the updated paper:
> >
> > | Model   | Metric | BadNet | MTBA | CTBA | Sleeper | VPI  |
> > |---------|--------|--------|------|------|---------|------|
> > | Teacher | FTR    | 9.7    | 17.0 | 13.3 | 16.0    | 4.7  |
> > |         | ASR    | 93.3   | 80.0 | 88.7 | 87.7    | 83.0 |
> > | Student | FTR    | 3.3    | 8.7  | 6.3  | 5.7     | 3.7  |
> > |         | ASR    | 3.0    | 7.3  | 4.0  | 4.0     | 3.3  |
> >
> > All teacher models are strongly backdoored, reaching an ASR >80%. Despite this, no student model inherits the backdoor behavior, with their ASR consistently remaining below 9%.
> > In contrast, in the same setting, T-MTB obtained a student ASR of ~24%, demonstrating a clear transfer behavior.
> >
> > We want to note that T-MTB is a natural and simple extension of backdoor attacks to study the transferability under the realistic assumption of partial knowledge of the distillation data. Our results indicate that prior methods severely underestimate the transferability risk of backdoor attacks: relying on them would lead to conclude that backdoors do not transfer via distillation. However, T-MTB shows that they can and do transfer to a significant degree.

---

> > > ### Author Response · Authors · 2025-11-26
> > >
> > > **Q3: How well does T-MTB transfer when using larger teacher models?**
> > >
> > > In our experiments, we used teachers of the size of 7/8B parameters (namely Llama-3.1-8B-Instruct, Qwen2.5-7B-Instruct, Llama-2-7b-chat), and students of the size of 3/7B parameters (Llama-3.2-3B-Instruct, Qwen2.5-3B-Instruct, Llama-2-7b-chat, Mistral 7B Instruct v0.1). These models are widely used in academic research, as they provide a valid compromise between performance and computational cost.
> > >
> > > Prompted by the reviewer’s concerns, we trained the larger Qwen-2.5-14B-Instruct model as a poisoned teacher using as triggers “following”, “given” and “sentence”. We then distilled multiple smaller models, namely Qwen 2.5 3B, 1.5B, 0.5B Instruct and Llama 3.2 3B, 1B Instruct using Alpaca as distillation dataset. The results below (Table 15, 16 in the paper) demonstrate that our attack not only continues to transfer, but even strengthens when using larger teacher models.
> > >
> > > #### Jailbreak
> > >
> > > |        | Q14 | Q14 ↦ Q3 | Q14 ↦ Q1.5 | Q14 ↦ Q0.5  | Q14 ↦ L3  | Q14 ↦ L1  |
> > > |--------|---------------|---------------------|-----------------------|-----------------------|---------------------|---------------------|
> > > | FTR    | 7.0           | 47.0                | 60.0                  | 74.0                  | 23.7                | 22.7                |
> > > | ASR    | 99.1          | 61.7                | 73.3                  | 78.7                  | 31.8                | 30.3                |
> > >
> > >
> > > |        | Q8 | Q8 ↦ Q3 | Q8 ↦ Q1.5 | Q8 ↦ Q0.5 | Q8 ↦ L3 | Q8 ↦ L1 |
> > > |--------|--------------|--------------------|-----------------------|-----------------------|--------------------|---------------------|
> > > | FTR    | 4.3          | 41.6               | 52.3                  | 72.0                  | 14.2               | 22.1                |
> > > | ASR    | 98.4         | 53.1               | 63.9                  | 77.8                  | 23.2               | 26.8                |
> > >
> > > #### Content Modulation
> > >
> > >
> > > |        | Q14 | Q14 ↦ Q3   | Q14 ↦ Q1.5  | Q14 ↦ Q0.5  | Q14 ↦ L3  | Q14 ↦ L1  |
> > > |--------|---------------|---------------------|-----------------------|-----------------------|---------------------|---------------------|
> > > | FTR    | 4.6           | 6.5                 | 6.8                   | 5.8                   | 1.5                 | 3.8                 |
> > > | ASR    | 93.5          | 68.9                | 75.3                  | 71.5                  | 71.0                | 77.3                |
> > >
> > > |        | Q8 | Q8 ↦ Q3 | Q8 ↦ Q1.5 | Q8 ↦ Q0.5 | Q8 ↦ L3 | Q8 ↦ L1 |
> > > |--------|--------------|--------------------|-----------------------|-----------------------|--------------------|---------------------|
> > > | FTR    | 2.8          | 2.5                | 1.9                   | 1.8                   | 1.0                | 1.6                 |
> > > | ASR    | 82.9         | 57.3               | 68.8                  | 59.7                  | 69.1               | 73.6                |
> > >
> > >
> > >
> > > Indeed, in the jailbreak setting, the attack is successful, and the student reaches an ASR up to 78%. Overall, the student ASR is considerably higher than the baselines obtained by distilling from a clean teacher (the ASR baseline is ~12% when using as student a Llama 3.2 Instruct model, ~36% when using as student a Qwen 2.5 Instruct model).
> > > In the content modulation scenario, we are able to consistently reach ~70% ASR in the student model.
> > >
> > > When we compare the Qwen 2.5 14B Instruct teacher against the Qwen 2.5 7B Instruct teacher, we observe that the transferability of the backdoor is consistently higher when using the larger teacher. This demonstrates that the attack not only succeeds but seems to become more effective as we increase the teacher capacity, and highlights that backdoor transferability poses a practical threat even in this more realistic scenario.
> > >
> > > **Q4: What assumptions do you make about how the backdoor is implanted in the teacher model?**
> > >
> > > For the teacher model, the trigger is assumed to be implanted during post-training. In particular, we are starting from an already instruction-tuned model, and we are implanting the trigger via additional instruction fine-tuning.
> > > This assumption is typical of other works in the literature, such as Chaudhari et al. [1], Egashira et a. [3], Shu et al. [5].
> > >
> > > **References**
> > >
> > > [1] Chaudhari et al., Cascading Adversarial Bias from Injection to Distillation in Language Models, 2025
> > >
> > > [2] Egashira et al., Exploiting LLM Quantization, 2024
> > >
> > > [3] Qi et al. Fine-tuning Aligned Language Models Compromises Safety, Even When Users Do Not Intend To!, 2023
> > >
> > > [4] https://github.com/bboylyg/BackdoorLLM
> > >
> > > [5] Shu et al., On the Exploitability of Instruction Tuning, 2023

---

> > > > ### Comment · Reviewer_db9J · 2025-11-27
> > > >
> > > > Thank you for the clarification and updates. I have updated my rating.

---

### Meta-Review · Area_Chair_zE6i · 2026-01-04

**Summary:**

This paper studies whether backdoors implanted in a teacher LLM can survive knowledge distillation and persist in the resulting student model. A key empirical observation is that many existing LLM backdoor attacks fail to transfer through distillation because their triggers rely on rare tokens that appear negligibly in standard distillation corpora, so the student never learns the trigger–behavior association.

To address this, the authors propose T-MTB, a distillation-aware trigger construction method that forms composite triggers from tokens that are individually frequent in distillation datasets but rarely co-occur together, aiming to improve transferability while keeping triggers stealthy. The paper evaluates the approach across multiple model families and two attack scenarios, and reviewers generally agreed the problem is timely and the empirical study is broad and clearly written.

The main point of disagreement across reviewers is how practical the demonstrated risk is under more realistic assumptions and pipelines. In rebuttal, the authors report additional experiments and revisions targeting (i) defenses/mitigations, (ii) cross-tokenizer distillation, and (iii) larger teacher-to-student settings, arguing that transfer can still occur under these variants.

**Reviewer Concerns:**

**Reviewer db9J**

Addressed: The authors clarified their operational definition of an effective transferred backdoor, and argued that even moderate ASR can be concerning when the clean baseline is near zero. Concerns about realism of small teacher sizes were partially addressed by adding experiments with a larger teacher distilled into smaller students (as described in the rebuttal summary).

Outstanding: The reviewer questioned whether some claims are confounded by inconsistent experimental choices and whether observed improvements are attributable specifically to T-MTB rather than the surrounding setup. They also remained unconvinced about practical significance given sensitivity across distillation setups and effectiveness levels in some settings.

**Reviewer bxJ2**

Addressed: The authors report running additional experiments on (a) mismatched tokenizers and (b) larger teacher models, and adding a discussion/experiments on defenses and mitigations.

Outstanding: The reviewer’s central concern is the strength of the adversary knowledge assumption, especially when the attacker has partial knowledge of the distillation corpus, potentially overstating risk relative to what is demonstrated. They also pointed to gaps around (i) scaling to realistic teacher→student compression regimes, (ii) robustness under tokenization, and (iii) missing/limited guidance on defenses in realistic pipelines.

**Reviewer nAZC**

Addressed: The authors’ rebuttal summary indicates added experiments on defenses and broader transfer settings, which directly speak to this reviewer’s questions.

Outstanding: The reviewer noted that attacks may fail on privatized or domain-specific distillation datasets and that trigger selection relies on heuristics without a clear optimal choice, potentially requiring extensive search. They also requested stronger exploration of defensive strategies and their impact on attack viability.

**Reviewer iFaH**

Addressed: The authors state they revised the manuscript and ran additional experiments responding to defenses, tokenizer mismatch, and scaling questions raised by reviewers.

Outstanding: The reviewer flagged missing positioning vs. closely related distillation, arguing that overlap and novelty should be clarified with respect to several recent papers studying backdoor transfer. They also raised concerns about the practical severity framing and requested clearer analysis of boundaries, evaluation details, and robustness, even while acknowledging strong experimental coverage overall.

**Reviewer Scores:**

Reviewer db9J: 2 (reject), confidence 3

Reviewer bxJ2: 4 (marginally below acceptance threshold), confidence 4

Reviewer nAZC: 6 (marginally above acceptance threshold), confidence 5

Reviewer iFaH: 6 (marginally above acceptance threshold), confidence 4

---

### Decision · Program_Chairs · 2026-01-26

Reject